# Machine Learning with Scarce Data:
# Ejection Fraction Prediction Using PLAX View

**Zhiyuan Gao**                                                             ZGAO2@CALTECH.EDU
*California Institute of Technology, Pasadena, California, USA*

**Dominic Yurk**                                                           DOMINICYURK@GMAIL.COM
*Asari AI, San Francisco, California, USA*

**Yaser S. Abu-Mostafa**                                                   YASER@CALTECH.EDU
*California Institute of Technology, Pasadena, California, USA*

**Editors:** Accepted for publication at MIDL 2025

## Abstract

We developed a machine learning model to predict left ventricular ejection fraction (LVEF/EF) from parasternal long-axis (PLAX) echocardiographic videos. Because public datasets with labeled PLAX videos are virtually non-existent, our work focuses on an innovative data generation strategy to overcome this scarcity. By leveraging a time-based correlation between clinical notes and echocardiographic videos, combined with fine-tuning view classifiers and proxy labeling, we effectively created a large labeled PLAX dataset and achieved a mean absolute error (MAE) of 6.86%. Given that Apical four-chamber methods, the clinical standard, report MAE values of 6%-7% (Ouyang et al., 2020), our results demonstrate that EF estimation from PLAX views is both feasible and clinically relevant. This surpasses the performance of existing methods and provides a clinically useful solution for situations where apical views may not be feasible. The EF labels for PLAX videos, derived from publicly available datasets, are accessible at https://github.com/Jeffrey4899/PLAX_EF_Labels_202501(Gao et al., 2025).

**Keywords:** Scarce data, Ejection fraction, Echocardiography, Parasternal Long-Axis (PLAX), Video View Classification, Proxy Labeling, PhysioNet MIMIC Dataset

## 1. Introduction

Cardiovascular disease is the leading cause of mortality worldwide, responsible for over 18.6 million deaths annually (Tsao et al., 2023). Echocardiography is a critical non-invasive diagnostic tool, with ejection fraction (EF) being a key parameter for assessing heart function. Accurate EF estimation aids in diagnosing conditions like heart failure and cardiomyopathies. While Apical four-chamber (A4C) view echocardiography is standard for EF estimation, obtaining high-quality A4C views can be challenging. In contrast, the parasternal long-axis (PLAX) view is often easier to acquire (Rao, 2025). However, there is no standard procedure for calculating EF from PLAX views. Previous efforts to estimate EF from PLAX views have shown promising results but leave room for improvement. For example, the ExoAI reported a PLAX-specific mean absolute error (MAE) of 7.29%, though

algorithmic details were not disclosed (Vega et al., 2024). Another study employing a landmark detection network achieved an MAE of 8.45% (Goco et al., 2022). A reliable method for direct EF estimation using PLAX that surpasses these results would greatly benefit patients for whom A4C views are not feasible.

The lack of public datasets linking EF values with PLAX views creates a significant bottleneck for machine learning (ML) research in this domain. While existing mature models have been trained primarily on A4C view, they do not generalize well to PLAX due to substantial differences in anatomical orientation and visual features. This paper addresses the gap by generating a novel dataset of PLAX echocardiographic videos with corresponding EF labels and training an ML model for EF prediction. These EF labels, which are aligned with existing echocardiographic data, are made publicly available at https://github.com/Jeffrey4899/PLAX_EF_Labels_202501(Gao et al., 2025), enabling reproducibility and further research.

## 2. Dataset Generation and Model Training

*The key step of our methodology—and the major challenge—was creating a labeled dataset for PLAX from available data, despite the scarcity of publicly available PLAX-specific labels. Because PLAX images and EF labels are not routinely paired in existing repositories, we developed novel techniques to both identify PLAX views and assign approximate EF values, effectively circumventing the lack of direct ground-truth annotations.*

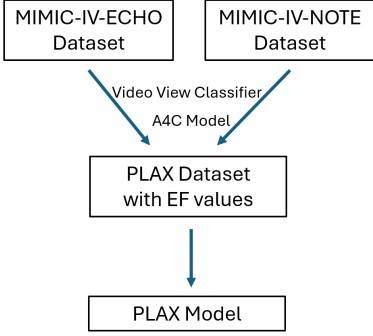

Figure 1: High-Level PLAX EF Prediction Pipeline.

Two major datasets from PhysioNet(Goldberger et al., 2000) were utilized for this study:

- **MIMIC-IV-Echo**(Gow et al., 2023): Contains approximately 500k echocardiographic videos without view type labels.

- **MIMIC-IV-Note**(Johnson et al., 2023): Includes around 331k discharge notes, of which only a subset contains EF values in unstructured text.

The goal was to extract PLAX view videos with EF data and use them to train a machine learning model for EF prediction. This task faced two main challenges: first,

the MIMIC-IV-Echo dataset lacked labels for echocardiographic view types, requiring the development of a video view classifier to identify PLAX views; second, the MIMIC-IV-Echo and MIMIC-IV-Note datasets were not directly linked, making it difficult to associate discharge notes with corresponding echocardiographic studies. Even after applying time-based correlations, the number of valid note-study pairs remained insufficient for training a robust PLAX model.

Consequently, we needed to leverage most studies in the MIMIC-IV-Echo dataset to generate training data for PLAX EF prediction. Specifically, we first trained a video view classifier to identify A4C and PLAX views within the dataset. Next, we trained an A4C model using a publicly available dataset to estimate EF values. These EF predictions were then applied as proxy labels for PLAX videos within the same study, enabling the development of a PLAX-specific model. A high-level overview is illustrated in Figure 1.

## 2.1. Video View Classifier Training

A classifier capable of distinguishing echocardiographic views into A4C, PLAX, and "OTHER," was critical for accurately selecting A4C and PLAX videos from the MIMIC-IV-Echo dataset.

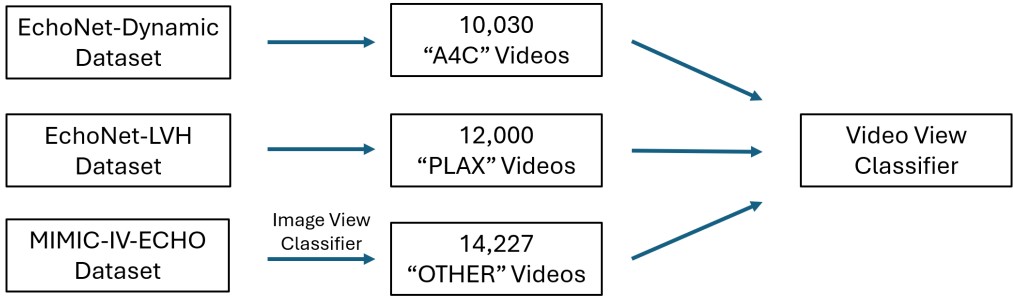

Figure 2: Multi-Dataset Strategy for Video View Classification.

To train such a classifier, we required videos labeled as A4C, PLAX, and "OTHER" views. This process is outlined in the flowchart in Figure 2.

EchoNet is a publicly available echocardiographic dataset designed for ML research. Unlike the MIMIC datasets, EchoNet datasets are highly curated collections of echocardiographic videos with specific annotations for view types and clinical parameters. For A4C and PLAX training data, we utilized the following two EchoNet datasets:

- **EchoNet-Dynamic**(Ouyang et al., 2020): Approximately 10,000 videos labeled as A4C with EF values.

- **EchoNet-LVH**(Duffy et al., 2022): Approximately 12,000 videos labeled as PLAX without EF values.

Generating a dataset for "OTHER" views was more challenging, as it required a wide variety of echocardiographic views to ensure the model could generalize effectively and identify the desired views with high accuracy.

To address this, we utilized the **TMED-2** dataset (Huang et al., 2022), which contains approximately 25,000 labeled echocardiographic images spanning various views, including A4C, PLAX, apical two-chamber (A2C), parasternal short-axis (PSAX), and a combined category of miscellaneous views labeled as "A4C/A2C/OTHER."

Using the labeled images from TMED-2, a ResNet-34 model was trained as an image classifier. This classifier was then applied frame-by-frame to videos in the MIMIC-IV-Echo dataset, and the aggregated frame-level predictions were used to assign video-level classifications. The final output of the model was a log-softmax output for each view class, and the exponential (exp) transform was applied to convert these log probabilities into meaningful scores. For a video, the class with the highest score was defined as model's view prediction.

Since we don't have the ground-truth labels for MIMIC-IV-Echo dataset, independently manual verification of 300 randomly selected test videos revealed the following:

- For 100 videos predicted as A4C, two reviewers identified 91/87 as correct.

- For 100 videos predicted as PLAX, two reviewers identified 78/72 as correct.

- For 100 videos not predicted as A4C/PLAX, two reviewers identified 94/98 as correct.

The classifier demonstrated high reliability in identifying "OTHER" views. Consequently, we utilized it to generate the "OTHER" dataset from MIMIC-IV-Echo dataset, defined as:

- 4,212 A2C videos (exp-transformed score $> 0.6$),

- 4,015 PSAX videos (exp-transformed score $> 0.6$),

- 6,000 videos labeled as "A4C/A2C/OTHER" (exp-transformed score $> 0.9$).

The exp-transformed scores were manually set to balance the number of videos across different categories, ensuring adequate representation for training.

First, for better model performance, it's desirable to balance A4C, PLAX, and "OTHER" categories, targeting "OTHER" dataset sizes similar to EchoNet-Dynamic (10,030 A4C) and EchoNet-LVH (12,000 PLAX).

Second, due to the limitations of the TMED dataset, the "OTHER" category was constructed using A2C, PSAX, and "A4C/A2C/OTHER" labels. Within "A4C/A2C/OTHER", manual verification showed that increasing the exp-transformed score threshold reduced A4C/A2C contamination. Thus, we set 0.9 to ensure diversity while minimizing A4C inclusion.

Third, to balance A2C and PSAX within "OTHER", while maintaining overall dataset proportionality between A4C, PLAX and "OTHER", we set a 0.6 threshold for both A2C and PSAX. This approach allowed us to create a comprehensive "OTHER" dataset while maintaining diversity in the video views.

Finally, the overall distribution of training data for the video view classifier was as follows:

- **A4C:** 10,030 videos, sourced entirely from the EchoNet-Dynamic dataset.

- **PLAX:** 12,000 videos, sourced entirely from the EchoNet-LVH dataset.

- **"OTHER":** 14,227 videos, ML-classified from the MIMIC-IV-Echo dataset.

For the final video view classifier, we utilized a pretrained X3D-s model.(Feichtenhofer, 2020) The model was trained using our labeled data. Compared to the previous image classifier, this model demonstrated significantly better performance in identifying A4C and PLAX videos, ensuring higher-quality data for subsequent analyses. This video-based classifier formed the backbone of our video view classification pipeline, achieving robust performance and enabling accurate selection of PLAX and A4C videos for downstream tasks.

### 2.2. A4C Model Training

The A4C model was trained using the **EchoNet-Dynamic** dataset, which contains 10,030 videos of A4C view with corresponding ejection fraction (EF) values. The methodology reported in (Ouyang et al., 2020) was followed. A 3D R(2+1)D convolutional neural network was implemented, achieving a mean absolute error (MAE) of 4.37% on the test set. This performance closely matches the MAE of 4.1% reported in the original study.

### 2.3. Ground Truth Data Generation

A ground truth dataset is essential for evaluating the true error of the PLAX EF model. Since the PLAX EF model is trained on EF values indirectly generated by the A4C model, its final error inherently includes the compounded error from the A4C model. To measure the real error of the PLAX EF model independently, a ground truth dataset with directly validated EF values is necessary. A brief overview of this process is illustrated in Figure 3.

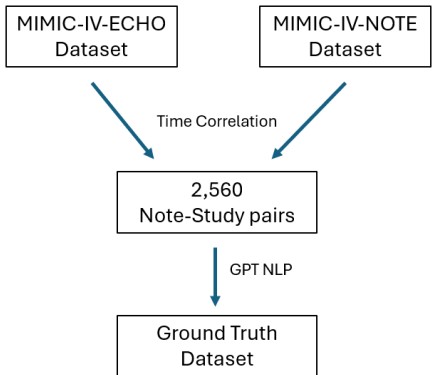

Figure 3: Ground Truth Dataset Generation Flow Chart.

No direct mapping exists between studies in the MIMIC-IV-Echo and MIMIC-IV-Note datasets. To address this, we performed a time-based correlation between the two datasets using the same patient ID. This approach identified 2,560 note-study pairs where echocardiography studies and clinical notes were recorded within a 60-day period.

Given the unstructured and messy text format of the notes, we utilized the GPT-4 API(OpenAI and et al., 2024) to extract key information, including EF values, from the discharge summaries. The API was instrumental in parsing and cleaning the free-text notes to retrieve meaningful clinical data, which was also publicly available (Gao et al., 2025). After further filtering the data (e.g., excluding studies with invalid EF data, color Doppler studies, and videos shorter than 2 seconds), we obtained 921 valid note-study pairs within the 60-day window.

To ensure these pairs contained A4C and PLAX view videos, we applied the video view classifier trained in Section 2.1. Among the 902 pairs, 848 were identified as having an exp-transformed score greater than 0.5 for both A4C and PLAX views. The EF values for these pairs were validated using the A4C model, yielding a MAE of 7.62%.

To improve accuracy further, we restricted the time correlation to a 1-day window, identifying 295 note-study pairs with an MAE of 6.64%. This error reflects the performance of EF labels generated by our A4C model compared with the note-extracted ones and closely matches the out-sample performance of 6.0% reported in (Ouyang et al., 2020). These 295 studies formed our ground truth test set and labels are publicly available(Gao et al., 2025).

Although this ground truth dataset is insufficient for training the PLAX model, it serves as an independent test set. The error from this test set provides a reliable evaluation of the PLAX EF model that does not depend on EF values generated by the A4C model.

## 2.4. View Classifier Fine-Tuning

To enhance the performance of our video view classifier (X3D model), we leveraged the 902 valid note-study pairs identified within the 60-day window as described in Section 2.3. These pairs were used to further refine the classifier's ability to identify A4C views.

First, we applied the view classifier to extract A4C videos from studies, identifying a total of 4,131 videos within the 60-day window. Next, we ran our A4C model on these identified A4C videos, producing the error distribution shown in Figure 4.

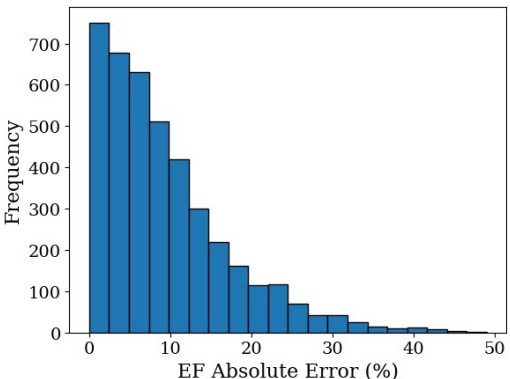

Figure 4: Error distribution of the A4C model within 60-day window.

Upon reviewing the 10 videos with the highest errors, we found that their views were either not true A4C views or partial A4C views. This observation demonstrated that the errors of the A4C model could be effectively utilized to fine-tune the video view classifier.

To address this, we fine-tuned the video view classifier using a subset of videos with extreme errors:

- Videos with MAE < 3% were labeled as A4C.

- Videos with MAE > 20% were labeled as "OTHER."

This subset comprised 1,346 videos, which were split equally into training and test sets for fine-tuning.

The X3D model was fine-tuned using the labeled training set, with the objective of improving its ability to distinguish between A4C and "OTHER" views. After fine-tuning, the test set was regenerated using the fine-tuned classifier, and the MAE was recalculated using the A4C model. The MAE was reduced from 6.83% to 5.14%, demonstrating a significant improvement in classifier accuracy.

This fine-tuning process enabled the classifier to more accurately identify A4C views, reducing misclassifications and ensuring higher-quality input for downstream tasks.

### 2.5. PLAX Dataset Generation

We applied the video view classifiers, both before and after fine-tuning, to the MIMIC-IV-Echo dataset, where the patients appearing in the ground truth dataset were excluded.

The label distribution of videos before fine-tuning is shown in Figure 5(a), while the distribution after fine-tuning is shown in Figure 5(b). After fine-tuning, the classifier identified A4C views more strictly, reducing the number of videos labeled as A4C by approximately 15,000, resulting in a total of around 20,000 videos. Conversely, the number of videos classified as PLAX views increased significantly.

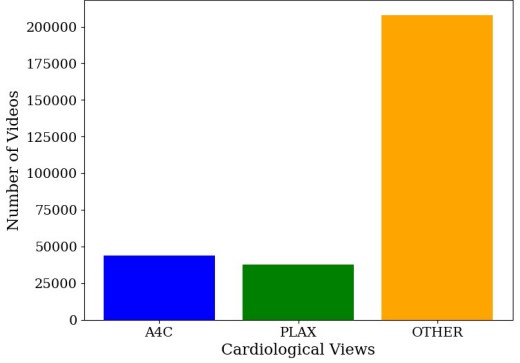
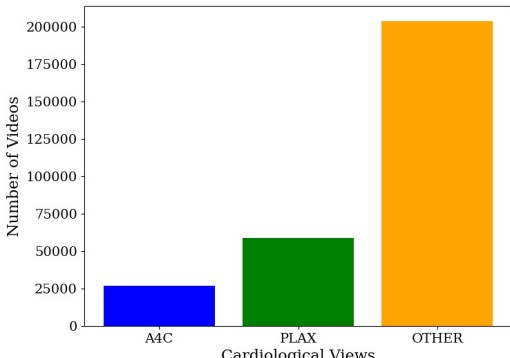

(a) Label distribution before fine-tuning.        (b) Label distribution after fine-tuning.

Figure 5: Comparison of label distributions before and after fine-tuning the video view classifier.

To ensure maximum accuracy, we selected PLAX views from the pre-fine-tuning classifier and A4C views from the post-fine-tuning classifier, eliminating any overlapping videos. Most studies contained both an A4C video and a PLAX video. For A4C videos in each study, we used the A4C model described in Section 2.2 to generate EF values, and then averaged those values to serve as the EF label for the PLAX videos in the study.

The final PLAX training dataset consisted of 25,532 videos, comprised 4822 studies (80% training, 20% validation). Different videos within the same study were assigned to the same split for a clean validation set. Labels are publicly available (Gao et al., 2025).

For testing, we used the ground truth dataset described in Section 2.3. Using the pre-fine-tuned view classifier, which is stricter in identifying PLAX views, we extracted 1,708 PLAX videos from the 295 studies. To ensure strict data separation, patients included in the test set were excluded from both the training and validation sets.

## 2.6. PLAX Model Training and Results

We trained a series of X3D and R(2+1)D models with various configurations, including different Batch Sizes, and Video Resolutions. The training details and configurations are summarized in Table 1.

Table 1: Training configurations for PLAX EF prediction models.

| Model | Batch Size | Resolution | MAE | Percentage |
|---|---|---|---|---|
| R(2+1)D | 16 | $112 \times 112$ | 6.98% | 20% |
| R(2+1)D | 32 | $112 \times 112$ | 7.03% | 20% |
| X3D-s | 12 | $224 \times 224$ | 7.03% | 20% |
| X3D-m | 8 | $224 \times 224$ | 6.90% | 40% |

All models were trained for 100 epochs using a learning rate (LR) scheduler with an initial LR of 0.001, a patience of 5 epochs and a reduction factor of 0.1. We applied preprocessing steps including padding and random cropping to the input videos before training. For each model, the epoch checkpoint with the lowest validation error was selected as the final model for evaluation. During testing, for each study, the predicted EF value is obtained by averaging the predictions of all its videos. The MAE is then computed at the study level.

The final EF prediction was obtained by linearly combining the outputs of the four models in Table 1, with ensemble weights manually tuned on the test set. After several experimental trials, the final weight distribution yielding the best performance was chosen. This best-performing ensemble achieved a final **MAE of 6.86%**.

To further assess the agreement between predicted and true EF values, we computed the Pearson correlation coefficient of 0.659, indicating a reliably positive correlation between our model's predictions and the ground truth. Additionally, we present a Bland-Altman plot (Figure 6) to analyze systematic bias and agreement limits. The mean difference (bias) is -0.22%, suggesting no significant systematic offset in predictions. The upper and lower limits of agreement (LoA) are 17.27% and -17.70%, respectively, indicating some variability in the prediction errors. Notably, the plot shows increased dispersion at higher EF values, which aligns with previous observations in EF estimation models.

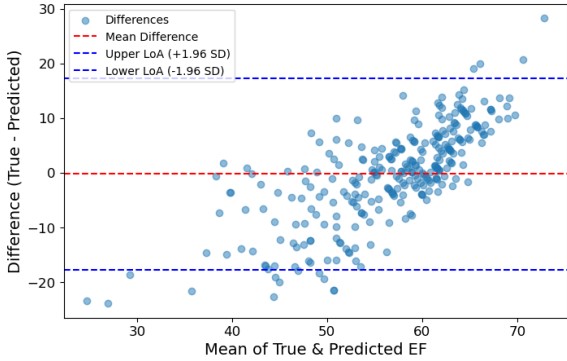

Figure 6: Bland-Altman Plot.

One potential explanation for this variance and the moderate correlation is the use of proxy labels derived from an A4C-based EF model. Since the A4C model was not trained on PLAX views, its predictions inherently introduce some domain shift and compounded errors, which may limit the upper bound of achievable performance. However, despite these challenges, our work establishes the **first benchmark for PLAX EF prediction** with disclosed algorithmic details and surpasses all previously published approaches. Prior works either lack methodological transparency (e.g., ExoAI with an MAE of 7.29%) (Vega et al., 2024) or rely on indirect EF estimation methods such as LVID measurement (MAE 8.45%) (Goco et al., 2022), which introduce inter-observer variability. By directly predicting EF from full PLAX cine videos, our approach avoids these manual dependencies while setting a reproducible standard for future research.

## 3. Conclusion

In this study, we developed a novel ML pipeline to predict EF from PLAX videos, addressing the scarcity of labeled PLAX data by leveraging existing public datasets. Our approach incorporated robust video view classification and proxy labels derived from an A4C model, enabling large-scale training. While this introduced some domain shift and potential errors, our final model achieved a **MAE of 6.86%**, surpassing existing benchmarks and establishing **a reproducible standard** for PLAX EF estimation.

To further improve and validate our model, we are initiating a collaboration with a leading heart hospital to leverage their clinical datasets for refining label accuracy and enhancing generalizability. Future work will focus on incorporating expert-annotated PLAX EF values, for which a dataset is currently being secured. Additionally, we plan to evaluate clinical applicability through external validation and potential clinical trials, ensuring real-world effectiveness in echocardiographic workflows.

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
