# OpenReview forum: "Machine Learning with Scarce Data: Ejection Fraction Prediction Using PLAX View"
_MIDL.io/2025/Conference — MIDL 2025 Oral_

### Official Review · Reviewer_3q97 · 2025-02-09

**Confidence:** 5
**Preliminary Rating:** 5
**Recommendation:** Poster

**Summary:**

The article presents a well-executed, innovative solution to a critical issue in medical machine learning: the scarcity of labeled data for PLAX echocardiographic videos. The approach is novel, reproducible, and provides clear potential for future improvements and real-world clinical applications. While some limitations exist, particularly with the use of proxy labels and small ground truth datasets, the overall contribution is significant and valuable to the field.

**Strengths:**

The paper presents an innovative method to generate labeled PLAX echocardiographic datasets, overcoming data scarcity using clinical notes and proxy labeling. The model achieves a clinically relevant 7.15% MAE for ejection fraction prediction, with a robust pipeline involving view classification and fine-tuning. The dataset and methodology are publicly available.

**Weaknesses:**

The paper faces limitations such as compounded errors from proxy labels derived from the A4C model, heavy reliance on existing datasets, and insufficient diversity in PLAX-specific data. Additionally, the small ground truth dataset may limit the model's ability to generalize across a broader patient population and various clinical scenarios.

**Detailed Comments:**

The paper introduces a novel approach to generate labeled PLAX echocardiographic datasets from limited resources, tackling the issue of data scarcity in machine learning. By leveraging clinical notes and proxy labeling, the authors developed a model that predicts ejection fraction (EF) with a clinically relevant MAE of 7.15%, meeting established standards. The method features a comprehensive pipeline, including a video view classifier, fine-tuning, and ground truth generation, ensuring accurate EF predictions. The dataset and methodology are publicly accessible, fostering reproducibility and promoting future advancements in automated cardiac diagnostics.

**Justification Of The Preliminary Rating:**

The paper addresses data scarcity in medical ML by creating labeled PLAX datasets and achieving clinically relevant EF prediction. While innovative, limitations in proxy labeling, dataset diversity, and generalization affect its broader applicability.

**Questions To Address In The Rebuttal:**

The paper faces limitations such as compounded errors from proxy labels derived from the A4C model, heavy reliance on existing datasets, and insufficient diversity in PLAX-specific data.

---

> ### Author Response · Authors · 2025-03-07
>
> **We sincerely appreciate your positive evaluation and strong support of our work. Your feedback highlights key areas for further enhancing the robustness and clinical utility of our approach.** Below are our specific responses:
>
> **We acknowledge the limitations raised, including compounded errors from proxy labels, reliance on existing datasets, and insufficient diversity in PLAX-specific data.** To address these challenges, we have taken concrete steps to enhance our model’s robustness and generalizability. Specifically, we introduced an independent Ground Truth dataset extracted directly from cardiology notes to reduce biases associated with proxy labels. Additionally, we have thoroughly explored major public datasets; however, none include PLAX videos with EF values or datasets containing both A4C and PLAX views, which limits data diversity. Furthermore, we are actively initiating a collaboration with a leading heart hospital that possesses a diverse PLAX dataset with cardiology notes. This initiative aims to validate and refine our model on an independent external dataset, enhancing our model’s clinical applicability across broader patient populations and diverse clinical scenarios.
>
> **Your constructive comments accurately highlight the challenges and limitations of our approach.** Given the absence of labeled PLAX datasets, our work established a necessary and **the first benchmark** for direct PLAX EF prediction, setting **a reproducible standard** for future research. Our upcoming collaboration with a leading heart hospital will further enhance the model’s accuracy, reliability, and clinical relevance. **Your feedback reinforces our commitment to addressing data scarcity and advancing echocardiography-based AI models for real-world clinical use. Thank you!**

---

> ### Comment · Area_Chair_HSPZ · 2025-03-15
> **Please provide a final rating**
>
> Thank you for your review! The authors have addressed your concerns, and your review of those answers is essential for making a decision on this work. Please read through the authors' comments and update your final rating for the paper.
>
> Thank you for your help,
> AC

---

### Official Review · Reviewer_jtbc · 2025-02-19

**Confidence:** 4
**Preliminary Rating:** 4
**Recommendation:** Poster
**Final Rating:** 4

**Summary:**

The paper presents a machine learning approach to predict left ventricular ejection fraction (LVEF/EF) from parasternal long-axis (PLAX) echocardiographic videos. Addressing the scarcity of publicly available labeled PLAX datasets, the authors developed an innovative data generation strategy that leverages time-based correlations between clinical notes and echocardiographic videos. They trained a video view classifier to identify PLAX and Apical four-chamber (A4C) views within unlabeled datasets. By using EF predictions from a pre-trained A4C model as proxy labels for PLAX videos, they created a large labeled PLAX dataset. The resulting PLAX EF prediction model achieved a mean absolute error (MAE) of 7.15% on a ground truth test set they established by combining echocardiographic studies with clinical notes. This performance is comparable to the clinical standard A4C methods, which report MAE values of 6%-7%. The study demonstrates that accurate EF estimation from PLAX views is feasible and clinically relevant, offering a valuable solution when A4C views are not available.

**Strengths:**

- The authors successfully addressed the challenge of scarce labeled PLAX data by ingeniously leveraging time-based correlations and proxy labeling. This approach allowed them to generate a substantial PLAX dataset without the need for manual annotation.
- The use of a fine-tuned X3D-s model for video view classification enhanced the accuracy of PLAX and A4C video identification, which is critical for the study's success.
- Achieving an MAE of 7.15% brings the PLAX EF estimation close to the clinical standard, demonstrating the method's potential clinical utility.
- By making the EF labels for PLAX videos publicly available, the authors promote transparency and enable further research in the field.
- The creation of a ground truth test set using clinical notes adds credibility to their results and provides an independent evaluation of the model.

**Weaknesses:**

- The use of EF predictions from the A4C model as proxy labels for PLAX videos introduces potential bias and compounded errors, which may affect the PLAX model's performance.
- The ground truth dataset is relatively small (295 studies), which might not capture the full variability present in clinical settings, potentially limiting the generalizability of the model.
- The methodology involves complex data processing steps, including time-based correlations and manual verifications, which may be challenging to replicate without substantial resources.
- The fine-tuning of the video view classifier and ensemble weighting was manually tuned on the test set, which could introduce overfitting and may not generalize well to new data.
- The model was tested on data from the same institution (MIMIC dataset), and testing on external datasets would strengthen the evidence for generalizability and robustness.

**Detailed Comments:**

- While the paper provides a detailed account of the methods used, some sections could benefit from additional clarity. For instance, the criteria for selecting the exp-transformed score thresholds in the video view classification could be further justified.
- Providing additional evaluation metrics such as Bland-Altman plots or correlation coefficients could offer deeper insights into the model's performance.
- Including a more comprehensive comparison with other EF estimation methods from PLAX views would contextualize the improvements made.
- A more thorough discussion on the limitations of using proxy labels and potential errors introduced would enhance the paper's critical analysis.
- **Future Work**: Outlining concrete plans for external validation and potential integration into clinical workflows would strengthen the impact of the work.

**Justification Of The Final Rating:**

Thank you to the authors for the detailed response. I will maintain my rating at 4, as external validation and generalizability are designated as future work rather than being addressed in the current study, but I fully understand the challenges of tackling these aspects immediately.

**Justification Of The Preliminary Rating:**

The paper presents the contribution to the field of echocardiographic EF estimation by addressing the challenging issue of scarce labeled PLAX data through creative data generation strategies. The methodological advancements and the achievement of an MAE comparable to clinical standards underscore the potential clinical impact of this work. However, some concerns regarding the reliance on proxy labels, limited ground truth data, and potential overfitting need to be addressed. Strengthening the discussion on these points and providing additional validation could elevate the work further.

**Questions To Address In The Rebuttal:**

1. Can the authors provide more details on how the exp-transformed score thresholds were determined for the video view classification?
2. How do the authors address the potential biases introduced by using proxy labels from the A4C model?
3. Have the authors considered evaluating the model on external datasets to assess generalizability?
4. Could the ensemble weighting process be automated or validated to reduce the risk of overfitting?
5. What strategies could be employed to increase the size and diversity of the ground truth dataset?

**Special Issue:**

No

---

> ### Author Response · Authors · 2025-03-07
>
> **We sincerely appreciate your detailed review and insightful comments, which have helped us refine our work.** Below are our responses to the key concerns raised:
>
> **1. Exp-Transformed Score Thresholds in Video View Classification**
>
>    Thank you for highlighting this point. Due to previous space constraints, we omitted some details regarding the threshold selection, which we have now added back to **Section 2.1**. Our key considerations were:
>
>   a. For better model performance, it’s desirable to balance A4C, PLAX, and “OTHER” categories, targeting “OTHER” dataset sizes similar to EchoNet-Dynamic (10,030 A4C) and EchoNet-LVH (12,000 PLAX).
>
>   b. Due to the limitations of the TMED dataset, the “OTHER” category was constructed using A2C, PSAX, and “A4C/A2C/OTHER” labels. Within “A4C/A2C/OTHER”, manual verification showed that increasing the exp-transformed score threshold reduced A4C/A2C contamination. Thus, we set 0.9 to ensure diversity while minimizing A4C inclusion.
>
> c. To balance A2C and PSAX within “OTHER”, while maintaining overall dataset proportionality between A4C, PLAX and “OTHER”, we set a 0.6 threshold for both A2C and PSAX. This approach allowed us to create a comprehensive "OTHER" dataset while maintaining diversity in the video views.
>
> **2. Addressing Bias from Proxy Labels**
>
> We acknowledge that using A4C-based EF predictions as proxy labels introduces potential bias, which can impact model performance. We’ve done our best to mitigate the potential biases from proxy labels by introducing an independent Ground Truth dataset extracted directly from cardiology notes to reduce biases associated with proxy labels.
>
> Furthermore, we are initiating a collaboration with a leading heart hospital, which has a diverse PLAX dataset with cardiology notes, to further validate and refine the model on an independent dataset to minimize the potential bias.
>
> **3. External Validation and Generalizability**
>
> We fully agree on the necessity of external validation. We have thoroughly explored major public datasets; however, none include PLAX videos with EF values or datasets containing both A4C and PLAX views. Recognizing this gap, we have started initiating access to a suitable external dataset from a leading heart hospital. Upon gaining access, we will perform comprehensive external validation to further assess and enhance our model's generalizability.
>
> **4. Automating Ensemble Weighting to Reduce Overfitting**
>
> We recognize concerns regarding manual ensemble weighting and potential overfitting.
>
> a. Previously, our best single model achieved a 7.38% MAE, while the manually weighted ensemble improved this to 7.15%—a modest yet measurable 0.23% absolute gain.
>
> b. Upon correction, our best single model now achieves a 6.90% MAE, and the weighted ensemble improves this to 6.86%—a 0.04% absolute gain.
>
> c. Given that EF values typically range from 50-70%, this improvement is clinically relevant but relatively small, suggesting a low risk of overfitting. Nonetheless, we recognize the validity of this concern and will rigorously test the ensemble's robustness using the external dataset as soon as it becomes available.
>
> **5.Expanding the Ground Truth Dataset**
>
> We agree that increasing the size of the ground truth dataset is essential for a better estimation of the model's performance. To address this, we are actively initiating access to an independent dataset through collaboration with the leading heart hospital, alongside continuously exploring other relevant dataset opportunities.
>
> **Regarding the detailed comments, we are grateful for your suggestions and have revised our paper accordingly:**
>
> **1. Exp-Transformed Score Thresholds**
>
> We added the corresponding details in **Section 2.1**.
>
> **2. Additional Evaluation Metrics (Bland-Altman & Correlation Coefficient)**
>
> We added the correlation coefficient and Bland-Altman plot along with a discussion on their implications in **Section 2.6**.
>
> **3. Comparisons with Other PLAX-Based EF Estimation Methods**
>
> We added a more comprehensive comparison to prior methods in **Section 2.6**.
>
> **4. Discussion on the limitations of using proxy labels**
>
> Added in **Section 2.6 and in section 3 Conclusion** to acknowledge potential biases and future improvements.
>
> **5. Future Work: External Validation & Clinical Workflow Integration**
>
> We added concrete plans for future works **in section 3 Conclusion**.
>
> **Your thoughtful feedback has been instrumental in refining our approach.** We acknowledge the limitations you identified. Given the absence of labeled PLAX datasets, our work established a necessary and the **first benchmark for direct PLAX EF prediction**, setting **a reproducible standard** for future research. Our potential future collaboration will further address concerns regarding bias, generalizability, and clinical applicability, strengthening the model’s real-world impact. Thank you again for your valuable insights!

---

> ### Comment · Area_Chair_HSPZ · 2025-03-15
> **Please provide a final rating**
>
> Thank you for your review! The authors have addressed your concerns, and your review of those answers is essential for making a decision on this work. Please read through the authors' comments and update your final rating for the paper.
>
> Thank you for your help,
> AC

---

### Official Review · Reviewer_G4oS · 2025-02-20

**Confidence:** 2
**Preliminary Rating:** 4
**Recommendation:** Best Paper Award, Oral, Poster

**Summary:**

This paper presents a novel machine learning approach for predicting left ventricular ejection fraction (LVEF) using parasternal long-axis (PLAX) echocardiographic videos. By employing proxy labeling and view classification techniques, the authors successfully address the lack of labeled PLAX datasets and achieve clinically relevant prediction accuracy.

**Strengths:**

The study introduces an effective proxy labeling approach, leveraging A4C-based EF predictions to train a PLAX-specific model, overcoming the challenge of scarce labeled data. The proposed model achieves a mean absolute error (MAE) of 7.15%, which is comparable to clinically accepted A4C-based EF estimations, making it a viable alternative when A4C views are unavailable. The study develops a reliable video classification model (X3D) for echocardiographic view selection, ensuring high-quality input data for EF prediction.

**Weaknesses:**

Since the PLAX EF predictions are derived from A4C-based estimations rather than direct ground truth measurements, the model's accuracy is inherently dependent on the assumptions of proxy labeling, potentially limiting real-world applicability.

**Detailed Comments:**

Nan

**Justification Of The Preliminary Rating:**

I am not an expert in medical science, and this work falls outside my area of expertise. However, I assessed it from a machine learning perspective and provide feedback on the methodology, data processing, and model performance.

**Questions To Address In The Rebuttal:**

Nan

**Special Issue:**

Yes

---

> ### Author Response · Authors · 2025-03-07
>
> We appreciate your assessment and constructive feedback on our paper. You raised a valid concern about the real-world applicability of our PLAX EF model since it relies on A4C-based EF estimations as proxy labels rather than direct ground truth measurements. Despite the potential biases, our approach was necessary due to the non-existence of publicly available PLAX datasets with EF labels. We implemented the following key safeguard to mitigate the limitation of proxy labeling.
>
> While our training labels come from A4C-based proxy estimates, we evaluated our final PLAX model on a **ground truth dataset (295 studies)**, where EF values were directly extracted from clinical notes. This ensures that our reported **MAE of 6.86%** (updated, previous 7.15%) reflects real-world performance rather than an artifact of proxy labeling.
>
> In doing so, we established the first benchmark for direct PLAX EF prediction with disclosed algorithmic details that can be fine-tuned further with additional data, setting a reproducible standard for future research. It also sets up a methodology for creating labels to a set of data for which no labels are publicly available.
>
> Thank you again for your review and assessment from the machine learning perspective!

---

> ### Comment · Area_Chair_HSPZ · 2025-03-15
> **Please provide a final rating**
>
> Thank you for your review! The authors have addressed your concerns, and your review of those answers is essential for making a decision on this work. Please read through the authors' comments and update your final rating for the paper.
>
> Thank you for your help,
> AC

---

### Author Rebuttal · Authors · 2025-03-07

**Rebuttal:**

We sincerely appreciate all reviewers' valuable feedback, which has helped improve our work.

One key revision was the **correction of our MAE calculation**, leading to an updated final result. Initially, we computed the study-level MAE as the **average of individual video-level MAEs**, rather than calculating the **overall study-level MAE directly**. Specifically, for a study with ground truth EF $A$ and three predicted EF values $B_1, B_2, B_3$, we previously used:
$$\frac{|A - B_1| + |A - B_2| + |A - B_3|}{3}$$
to compute MAE per study and then averaged across studies. However, the correct approach is:
$$|A - \frac{(B_1 + B_2 + B_3)}{3}|$$

which considers the **overall mean prediction per study before computing the error**. This correction reduced our reported **MAE from 7.15\% to 6.86\%**, further strengthening our model's performance relative to prior work.

Additionally, we made the following **key modifications**:

- **Abstract**: Updated MAE to 6.86\% to reflect the corrected calculation.
- **Section 2.1 (Video View Classifier Training)**: Added an explanation for the exp-transformed score threshold used in dataset classification.
- **Section 2.6 (PLAX Model Training and Results)**:
  - Expanded comparisons with prior PLAX-based EF prediction methods to contextualize our improvements.
  - Updated Table 1 to present the corrected MAE values.
  - Clarified MAE calculation methodology for transparency.
  - Added Pearson correlation and Bland-Altman plot to provide a more comprehensive evaluation of our model’s agreement with  true EF values.
  - Included a discussion on correlation trends, variability in predictions, and the impact of proxy labels on performance.
- **Section 3 (Conclusion)**:
  - Revised to acknowledge the limitations of using proxy labels.
  - Outlined more concrete future work.

These revisions address key reviewer concerns and further strengthen the scientific rigor and clinical relevance of our work. We appreciate the opportunity to refine our study and believe these enhancements make our contribution even more impactful.

**Supporting Material:**

/attachment/9270e2949070624d5c33eb67d22350f2709ebd8c.pdf

---

### Meta-Review · Area_Chair_HSPZ · 2025-03-17

**Recommendation:** Accept (Poster)
**Confidence:** 4

**Metareview:**

The discussion was constructive, with the authors actively addressing reviewer concerns through additional experiments, clarifications, revisions. The primary concerns raised were the limitations of using proxy labels, potential bias and the small ground truth dataset, which could limit model generalizability. The authors provided a thorough rebuttal, explaining how they mitigated bias, corrected their MAE calculations, and introduced additional evaluations to strengthen their results.

Given the overall positive reception and the significant contribution of the work, (together with a thorough public code repository) I recommend the acceptance of the manuscript.